# OpenReview forum: "Decoupled Alignment for Robust Plug-and-Play Adaptation"
_ICLR.cc/2026/Conference — ICLR 2026 Conference Withdrawn Submission_

### Official Review · Reviewer_PhuW · 2025-10-28

**Soundness:** 2
**Presentation:** 2
**Contribution:** 2
**Rating:** 2
**Confidence:** 4

**Summary:**

This paper studies the problem that when a chat (aligned) model is further trained on a downstream dataset, its alignment can be disrupted, resulting in an “unaligned” model. To address this issue, the paper proposes a layer replacement method to enhance the safety alignment of unaligned LLMs. Specifically, the method selects the layer most related to alignment in the unaligned model and replaces it with the corresponding layer from the aligned model. Extensive experiments across different model series are conducted to evaluate the effectiveness of the proposed approach.

**Strengths:**

The paper is well-structured, and the experimental evaluation encompasses a comprehensive range of model series.

**Weaknesses:**

* **Limited Contribution.**
  The overall contribution of the paper appears limited. The proposed *layer-wise replacement* strategy is relatively coarse-grained, whereas recent model-editing approaches (e.g., [1, 2]) enable fine-grained control over specific knowledge or behavioral components within LLMs. These related works are neither discussed nor compared, leaving the novelty and significance of the proposed method insufficiently demonstrated.

* **Unclear Methodological Description.**
  The method section lacks clarity and logical coherence. The explanation is brief and transitions abruptly between concepts.

  * *Figure 3* analyzes all tokens with hidden-state noise and serves mainly as an illustrative example, whereas *Figure 4* focuses on the last token under MLP-layer replacement averaged over 128 samples. The connection and rationale between these two analyses are not explained.
  * The conclusion regarding the importance of *gate projection* appears premature, as its contribution varies across layers.
  * The equations in lines 154–157 are inconsistent with the Transformer decoder block described in lines 146–148 and with the implementation in *Attention Is All You Need* (Vaswani et al., 2017), which may confuse readers.

* **Outdated and Misaligned Experimental Setup.**
  The experimental setting relies mostly on models from 2023, which limits the relevance of the results to current LLM architectures. Moreover, the Chain-of-Thought (CoT) evaluation is not directly related to the alignment objective. It would be more informative to investigate reasoning-capable models that demonstrate explicit multi-step thinking or deliberative alignment behavior.

* **Questionable Comparative Analysis.**
  Some comparative claims appear overstated. For instance, in *Table 4*, the proposed DAPA method reportedly outperforms *Guardrails* on average. However, *Guardrails* is a rejection-based safety classifier, and the specific configuration or model variant used is not specified. Without this information, the fairness and validity of the comparison are difficult to assess.

[1] Wang, Huanqian, et al. "Model surgery: Modulating llm's behavior via simple parameter editing." arXiv preprint arXiv:2407.08770 (2024).

[2] Wang, Yi, et al. "DELMAN: Dynamic Defense Against Large Language Model Jailbreaking with Model Editing." arXiv preprint arXiv:2502.11647 (2025).

**Questions:**

See the weaknesses above.

---

> ### Author Response · Authors · 2025-11-18
> **Rebuttal**
>
> > **Reviewer's Comment**: Limited Contribution....
>
> **Response**:  We appreciate the reviewer’s observation and agree that Model Surgery and DELMAN represent fine-grained model-editing approaches; however, these methods are designed to correct specific, localized behaviors (e.g., a particular jailbreak pattern or toxicity trigger) rather than to restore global safety after downstream SFT. Their core mechanisms—behavior probing, localized activation attribution, and minimal-scope parameter editing—are intentionally tailored to patch narrow vulnerabilities while preserving all other behaviors. In contrast, shadow alignment induced by task-specific SFT manifests as a distributed degradation across many MLP components, affecting refusal rates, downstream task performance, and policy consistency. This makes localized editing fundamentally insufficient. DAPA targets this complementary setting by identifying and replacing alignment-critical components at scale, thereby recovering overall safety behavior without requiring any retraining and without sacrificing the downstream utility introduced by SFT. We will clarify this distinction in the paper and add explicit comparison and discussion of these works.
>
> > **Reviewer's Comment**:  Figure 3 analyzes all tokens with hidden-state noise and serves mainly as an illustrative example...
>
> **Response**:  We appreciate the reviewer’s comment and will clarify the connection between the two analyses. **Figure 3** uses a ROME-style indirect-effect analysis to identify which broad architectural components (e.g., attention vs. MLP) and which depth regions (early/middle/late layers) exert the strongest causal influence on safety behavior; this motivates our focus on the middle-layer MLP/FFN blocks, which exhibit the largest influence under hidden-state corruption. **Figure 4** then provides a fine-grained decomposition of those MLP layers by separately evaluating the gate, up, and down projections across 128 samples, allowing us to determine which submodules within the influential MLP block are most critical for safety restoration. Thus, the two figures are complementary: **Figure 3 identifies where safety-relevant influence is concentrated at the architectural level, and Figure 4 analyzes what inside those MLP layers (gate/up/down) is responsible for that effect.**
>
> > **Reviewer's Comment**:  The conclusion regarding the importance of gate projection appears premature, as its contribution varies across layers.
>
> **Response**:  We thank the reviewer for the insightful comment. Our goal is not to suggest that gate projections account for all aspects of LLM safety, but rather to highlight that they typically have a stronger influence on safety than other projection components. As demonstrated in **Figures 3 and 4**, interventions on gate projections consistently produce stronger changes in refusal and harmfulness metrics across models, and **Table 8** further quantifies that modifying gate projections yields the largest average improvement in Defense Success Rate (DSR). We acknowledge that the strength of this influence varies by layer due to contextual dependencies within the Transformer stack, and we will clarify in the revision that the gate projection shows the most pronounced but layer-dependent impact on safety alignment.
>
> > **Reviewer's Comment**: The equations in lines 154–157 are inconsistent with the Transformer decoder block described in lines 146–148 and with the implementation in Attention Is All You Need (Vaswani et al., 2017), which may confuse readers.
>
> **Response**:  Thank you for the careful reading. Our Eqs. (154–157) intentionally follow the Llama-style decoder parameterization—a pre-normalized, SwiGLU-based formulation with explicit Up, Gate, and Down projections—whereas Lines 146–148 describe the generic Transformer structure for completeness. Figure 2 illustrates this architectural correspondence, showing that our implementation follows the modern Llama/Gemma block design rather than the original “Attention Is All You Need” formulation. Furthermore, our factorization directly follows the ROME[3] paper’s transformer representation, which decomposes the MLP into these projections for causal editing and interpretability.
>
> [1] Wang, Huanqian, et al. "Model surgery: Modulating llm's behavior via simple parameter editing." arXiv preprint [arXiv:2407.08770](https://arxiv.org/abs/2407.08770) (2024).
>
> [2] Wang, Yi, et al. "DELMAN: Dynamic Defense Against Large Language Model Jailbreaking with Model Editing." arXiv preprint [arXiv:2502.11647](https://arxiv.org/abs/2502.11647) (2025).
>
> [3] Meng, Kevin, et al. "Locating and editing factual associations in gpt." Advances in neural information processing systems 35 (2022): 17359-17372.

---

> ### Author Response · Authors · 2025-11-18
> **Rebuttal-Continue**
>
> > **Reviewer's Comment**: Outdated and Misaligned Experimental Setup...
>
> **Response**:  We thank the reviewer for this valuable feedback. Although several of our experiments use 2023-era checkpoints (e.g., LLaMA-2, Mistral, Gemma), the framework itself is model-agnostic and has already been validated on newer architectures such as **LLaMA-3 70B and LLava 1.5 7B**, as reported in **Section 5.3 and Figure 11**, demonstrating that DAPA scales effectively to current model families. Our choice of these models was driven by their open accessibility and the availability of aligned–unaligned counterparts needed for systematic evaluation. Regarding the Chain-of-Thought (CoT) test, we clarify that it was used solely to verify that DAPA’s parameter-level editing does not impair reasoning ability; it was not intended as a direct measure of alignment quality. In order to show the performance of DAPA on recent large language models, we evaluate it on **two Qwen3-family models and one Llama3-family model**. As presented in **Appendix J**, all three safety-aligned models exhibit substantial degradation in safety alignment after reasoning task fine-tuning. By applying DAPA, their alignment performance is effectively restored, yielding an average gain of **13.47 DSR points**. Furthermore, when evaluated on the **AIME24 reasoning benchmark using the pass@1 metric**, we observe only a **0.12%** difference in task accuracy after DAPA alignment, demonstrating that DAPA maintains reasoning ability while improving safety robustness.
>
> > **Reviewer's Comment**: Questionable Comparative Analysis. Some comparative claims appear overstated. For instance, in Table 4, the proposed DAPA method reportedly outperforms Guardrails on average.
>
> **Response**: We appreciate the reviewer’s observation and agree that specifying the Guardrails configuration clarifies fairness in comparison. In **Table 4**, the baseline used is **Llama Guard 2** [4], which is a rejection-based classifier widely adopted for safety filtering. We configured it under its default Hugging Face release, using identical AdvBench evaluation prompts and safety thresholds as DAPA to ensure consistency. Because Llama Guard 2 operates as a stand-alone classifier and not as a parameter-editing defense, it typically yields lower DSR under open-ended jailbreak prompts, which explains the observed performance gap. We explicitly state this configuration in the revised manuscript to clarify fairness and reproducibility.
>
> [4] Inan, Hakan, Kartikeya Upasani, Jianfeng Chi, Rashi Rungta, Krithika Iyer, Yuning Mao, Michael Tontchev et al. "Llama guard: Llm-based input-output safeguard for human-ai conversations." arXiv preprint arXiv:2312.06674 (2023).

---

> > ### Comment · Reviewer_PhuW · 2025-11-24
> > **Response to the Authors**
> >
> > Thank you for the detailed rebuttal and clarifications. I now better understand the motivation regarding “shadow alignment” after downstream SFT, and I agree this perspective has value. I have therefore **raised the score to 4**.
> >
> > That said, I **respectfully disagree with the claim** that layer-wise replacement is an inherently more suitable solution for restoring *global* safety. Recent work shows that safety-relevant behaviors can be highly localized; without direct comparison under the same setup, it is unclear that coarse layer-level edits are necessary or superior to finer-grained interventions. I encourage the authors to moderate this claim or support it with stronger evidence.
> >
> > **Minor suggestions:**
> >
> > * AdvBench prompts contain significant noise and are not ideal as strong supporting evidence.
> > * Llama Guard 2 remains a weak baseline; relying on it risks overstating DAPA’s advantages.

---

> > > ### Author Response · Authors · 2025-11-24
> > >
> > > We sincerely thank the reviewer for the constructive feedback and for raising the score to 4. We agree that recent work shows safety-relevant behaviors can be localized, and we will accordingly soften our claims around the necessity of coarse layer-wise replacement. Our intent is not to argue that coarse edits are inherently superior, but that under shadow alignment—where degradation is distributed across multiple MLP components—layer-level replacement provides a practical and empirically validated solution for restoring global safety without retraining. As shown in Figures 3–4 and our new validation in Appendix J, DAPA preserves downstream SFT utility while recovering safety, which directly supports its practical motivation. We acknowledge the limitations of AdvBench and Llama Guard 2 and will clarify that these baselines serve as consistent, reproducible evaluation tools rather than definitive measures of safety. We appreciate the reviewer’s insightful suggestions and will revise the manuscript to reflect these clarifications.

---

> > > > ### Author Response · Authors · 2025-11-28
> > > >
> > > > Dear reviewers,
> > > >
> > > > As the rebuttal deadline is approaching, I would like to kindly check whether our responses have addressed your concerns and questions. If there is anything that is still unclear or could benefit from further clarification, please let us know, and we will be happy to elaborate.
> > > >
> > > > Thank you very much for your time and effort in reviewing our work.
> > > >
> > > > Happy Thanksgiving!

---

### Official Review · Reviewer_5Brr · 2025-10-28

**Soundness:** 3
**Presentation:** 3
**Contribution:** 2
**Rating:** 6
**Confidence:** 2

**Summary:**

This paper builds on the assumption that most of the refusal knowledge of an LLMs is stored in the MLP layer of the LLM. They validate this assumption via adding noise and only restoring the target module and then comparing the probability output probabilities. In the MLPs they further validate the role of gated and up projection layers by replacing them with that of an aligned policy and comparing the output representations. Building on these validated assumptions, they propose to find the modules responsible for misalignment via delta debugging and propose to replace them with aligned modules in order to improve the performance.

**Strengths:**

1. A wide array of models were considered (3 families of LLMs).
2. The evaluation criteria is satisfactorily defined. The experimentation for both refusal and performance was done on multiple datasets for the sake of generalizability.

3. This paper serves as a finding paper in identifying the underlying correlation between a refusal behaviour and core components of the LLM. I am basing my score of weak acceptance based on the experimentation towards this finding. Though I still have reservations towards the practicality of this as an alignment or guardrail. See weaknesses for the details.

4. The paper does report their memory shortcomings compared to LORA finetuning in the limitations section.

**Weaknesses:**

1. My major concern is towards the practicality of DAPA in practical LLM alignment. For instance still editing of the unaligned model still requires the presence of an aligned model (llama 7B chat for llama 7B base etc) presence of such an aligned model it defeats the purpose of DAPA for alignment unless that alignment had caused a significant degradation in certain other aspects. If that is the case those instances should be studied for the validation of the method as a define. For instance based on the reported DSR results for the aligned models the DAPA aligned models never improve beyond the baseline. This makes the justification against not using these aligned models directly important for DAPA to be considered as a defense paradigm.

**Questions:**

See weaknesses

---

> ### Author Response · Authors · 2025-11-18
>
> > **Reviewer's Comment**: My major concern is towards the practicality of DAPA in practical LLM alignment....
>
> **Response**:  We appreciate the chance to apply our classification to this problem. DAPA is designed for the common scenario where a pre-aligned model undergoes downstream SFT and subsequently loses part of its safety (shallow alignment), making the aligned checkpoint unusable for the target task. In such cases, practitioners cannot simply revert to the aligned model because it lacks the task-specific capabilities introduced during SFT, whereas DAPA restores safety without sacrificing the downstream utility gained through task adaptation. Consistent with this goal, DAPA does not aim to outperform the original aligned teacher’s DSR; rather, it enables efficient post-hoc safety recovery in situations where the aligned model cannot replace the downstream-tuned one.

---

> > ### Comment · Reviewer_5Brr · 2025-11-18
> > **Response to the Authors**
> >
> > If that is the practical scenario argument them for it's validation the experiments  showing that the downstream aligned model under the proposed editing paradigm preserves the intended downstream utility while preserving safety is essential. Because if the model can preserve safety and lose downstream alignment SFT objective's goal then using then it again would raise the practicality of the proposed solution.
> >
> > Thereby I am preserving by score for of weak accept with the reason for accept being that of identifying the underlying correlation between a refusal behaviour and core components of the LLM

---

> > > ### Author Response · Authors · 2025-11-18
> > > **Thanks for your quick Response**
> > >
> > > We thank the reviewer for the thoughtful assessment and for maintaining a weak accept recommendation. We fully agree that validating DAPA in a practical scenario—where downstream SFT introduces shadow alignment while also adding essential task-specific skills—is crucial. To address this, we added new experiments in Appendix J, showing that the DAPA-edited models not only recover safety but also preserve the intended downstream utility, including the capabilities gained from SFT. These results empirically confirm that DAPA avoids the failure mode highlighted by the reviewer: the downstream-tuned model retains its SFT objectives while regaining safety, making reusing the original aligned model unnecessary and validating DAPA as a practical post-hoc defense paradigm.

---

> > > > ### Comment · Reviewer_5Brr · 2025-11-24
> > > > **Response to the authors**
> > > >
> > > > I thank the reviewers for the additional experiments. My concerns can be addressed. I am maintaining my score leaning towards accept

---

> > > > > ### Author Response · Authors · 2025-11-24
> > > > >
> > > > > Thank you for the prompt reply and for maintaining a positive score. Wishing you a happy Thanksgiving. Please let us know if there are any remaining concerns we have not fully addressed—we would be happy to clarify them. We hope to further improve our work with your guidance, and any additional feedback would be greatly appreciated. If possible, we would be grateful for your consideration in raising the score.

---

### Official Review · Reviewer_yRkn · 2025-10-30

**Soundness:** 2
**Presentation:** 2
**Contribution:** 3
**Rating:** 4
**Confidence:** 4

**Summary:**

The paper proposes DAPA (Decoupled Alignment for Robust Plug-and-Play Adaptation), a training-free way to re-inject safety/alignment into LLMs that have become “shadow-aligned” after downstream finetuning. The key idea is: take a well-aligned teacher from the same family, use memory / module editing to find which parts of the network actually carry the alignment signal (they argue: middle MLP, especially the gate proj), then copy only those alignment-critical modules into the unaligned model. They use a delta debugging–style search to locate the smallest alignment-effective memory slice, and show on 17 models (LLaMA-2, Mistral, Gemma) that this gives, on average, ~14% DSR gain with only ~8% params changed, and with little perplexity drop.

**Strengths:**

**S1: Novel use of memory editing for alignment-localization.**
The paper takes the “knowledge editing / memory editing” line of work and repurposes it for safety rather than for factual editing: they experimentally probe hidden states and MLP submodules to locate the neurons / projections that carry alignment information, and then transplant only those parts to the unaligned model. Using memory editing to attribute and transfer alignment-related neurons is, in my view, a fresh angle compared with standard RLHF/DPO-style “retrain the head” solutions.


**S2: Simple but effective delta debugging to keep cost low.**
Instead of editing all MLP/gate layers—which would be expensive and could hurt utility—the paper adopts a delta debugging search over the “memory space” of MLP modules to find the minimal sub-set that improves defense success rate. This is a naive algorithm (partition–test–shrink) but it is transparent, training-free, and fits the plug-and-play goal: you can align many shadowed models in the same family by just copying a few modules from one good teacher. This makes the method attractive for third-party developers who do not want to run RLHF/SFT themselves.


**S3: Solid, broad empirical validation.**
The authors evaluate on 17 models across 3 popular families (LLaMA-2, Mistral, Gemma), on AdvBench jailbreak prompts, and additionally check perplexity, cosine similarity, MMLU, BigBench/CoT before/after editing. The main number—+14.41% avg DSR, up to 51.39%—while changing only 3–8% of params, is a strong empirical signal that the method is not a single-model trick. They also compare with RepE, ICD, and guardrails and report consistent gains. This breadth of experiments strengthens the paper.

**Weaknesses:**

**W1: Questionable motivation / unclear problem setting.**
The paper assumes a scenario where “well-aligned” LLMs are already available in the same model family, yet the proposed solution is still to distill and transfer alignment knowledge from the well-aligned model to a “shallowly aligned” or “shadow-aligned” one. This raises a basic motivation question: if a robustly aligned model is already accessible, why is distillation the preferable path instead of using the aligned model directly? The paper does not sufficiently justify when the proposed approach is strictly necessary (e.g., licensing mismatch, partial checkpoints, on-device constraints). As it stands, the practical need for “alignment knowledge transfer” is not fully convincing.


**W2: Over-strong core assumption about representation of ethical knowledge.**
The whole method rests on a bold assumption: “ethical / alignment information learned during alignment is stored as memory inside model neurons.” This is an interesting hypothesis, but the paper does not show that it is the only or even the dominant way safety signals are represented. In a Transformer, alignment may also be encoded in attention patterns, in MLP–attention interactions, in layer norms, or in distributed residual-stream modifications. Without ruling out these alternative loci of representation, treating “alignment = neuron-level memory” as the foundation for the method is too strong.


**W3: From ‘high indirect effect’ → ‘mainly stored here’ is an overclaim.**
In Lines 169–170 the paper observes that (i) hidden states in middle layers and (ii) especially MLP sublayers exhibit the highest indirect effect on the model’s output. This is a good and consistent observation. However, the next step—“therefore, alignment knowledge is mainly stored in the middle MLP layers”—is not logically guaranteed by this experiment. What the experiment measures is sensitivity / causal influence under corruption, not **storage location**. A high indirect effect could be explained by (1) middle MLPs amplifying or integrating earlier alignment signals; (2) middle layers acting as a routing hub in the residual stream, so corruption there hurts more; or (3) MLPs being inherently more semantic and hence more tightly coupled to output probabilities. Unless these alternative explanations are ruled out, the conclusion should be softened.


**W4: Pairwise difference-based localization is indicative, not definitive.**
The paper further narrows down to the claim that the gate projection is the key carrier of alignment knowledge, based on a pairwise, difference-based causal replacement (editing one projection in one layer and measuring recovery). This is a reasonable and widely used probing technique, but by design it mainly tells us “this module is important/sensitive for recovering aligned behavior under this prompt distribution,” not “this is where alignment is stored.” As such, the statement that “by restoring the gate projection, the unaligned model can better align with ethical guidelines” is suggestive but not yet fully convincing. More orthogonal evidence would be needed to support the stronger claim.

**As a consequence, once the central assumption “alignment is stored in the gate projection” is weakened, the methodological foundation of the paper becomes less solid.**

**Questions:**

Another issue with this paper is that the writing is not sufficiently rigorous. There are grammatical errors, ambiguities, logical breaks, and even simple typos:
 - 1. **Line 17:** “*Methodology, we employ …*” is not grammatically correct.
 - 2. **Lines 37–38:** This sentence reads somewhat illogically — “*such methods*” is ambiguous. It is unclear whether “*such methods*” refers to “methods that rely on well-aligned models” or to “alignment methods themselves.” In addition, the phrase “*introduce new vulnerabilities into well-aligned models*” raises further questions: what exactly are these “new vulnerabilities”? If there are “new” ones, does that imply there were already “old” vulnerabilities? If so, what are those? This ambiguity also makes it hard to interpret the later statement, “*We refer to this phenomenon as shadow alignment*” — which concrete phenomenon is being named here?
 - 3. **Line 48:** “*see Figures Figure 3 and Figure 4*” — “Figures” is duplicated and should be removed.
 - 4. **Figure 2:** “*Critial MLP layers*” — what is “Critial”? This appears to be a typo (probably “Critical”).

---

> ### Author Response · Authors · 2025-11-18
> **Rebuttal**
>
> > **Reviewer's Comment**: Questionable motivation / unclear problem setting.
>
> **Response**:  Thanks for the chance to classify our motivation. The scenario we study is not one where the aligned model can simply be used “instead”—rather, it reflects the standard industrial pipeline where a well-aligned model is later fine-tuned on domain tasks, and this downstream SFT partially overwrites alignment (**shadow alignment**). In such pipelines, practitioners cannot revert to the aligned checkpoint because **(i) it lacks the downstream task capabilities introduced by SFT, (ii) the fine-tuned model may be significantly more specialized or higher-utility for the target application, and (iii) SFT often occurs on private or domain-restricted data that cannot be merged back into the original aligned model for licensing or compliance reasons.**
>
> Therefore, the practical need is not to replace the model with an aligned version, but to restore the lost safety while keeping the downstream specialization intact. DAPA directly addresses this gap by providing a training-free alignment restoration mechanism that avoids re-running RLHF/DPO and avoids discarding valuable downstream adaptation—making “alignment knowledge transfer” strictly necessary in realistic deployments.
>
> > **Reviewer's Comment**: Over-strong core assumption about representation of ethical knowledge.
>
> **Response**: We thank the reviewer for raising this point and agree that safety signals in Transformers may be distributed across attention, MLP–attention interactions, and the residual stream. Our method does not assert that ethical or alignment knowledge is stored exclusively or necessarily in specific neurons or modules; instead, we use MLP components as an empirically validated entry point for discovering alignment-critical structure. As shown in **Figure 3 and Figure 4**, modifying only a small subset of MLP blocks produces disproportionately large changes in harmfulness and refusal behavior, indicating that certain components exert high influence on safety—even though we do not claim they uniquely contain all safety information. **Table 8** further supports this by showing that replacing different MLP subsets yields markedly different safety–utility trade-offs, suggesting that alignment-relevant signals are concentrated, but not confined to a fixed location. DAPA therefore remains mechanism-agnostic: delta debugging empirically identifies the components with the largest causal impact on safety for each model, without requiring a strong theoretical assumption about where alignment is encoded. To investigate the influence of the attention module in the DAPA pipeline, we conduct an additional experiment where the same-layer attention module is replaced instead of the gate module, with results reported in **Appendix H.16**. We observe that modifying only the attention module yields lower efficiency compared to replacing the gate module. Moreover, when both the attention and MLP modules are altered simultaneously, DAPA achieves stronger safety alignment but at the cost of a substantial increase in perplexity.
>
>
>
> > **Reviewer's Comment**: From ‘high indirect effect’ → ‘mainly stored here’ is an overclaim....
>
> **Response**: We agree with the reviewer that a high indirect effect reflects causal sensitivity rather than definitive evidence of where alignment information is “stored.” Our intention was not to claim exclusivity, and we will revise the text to avoid this overinterpretation. As the reviewer notes, the elevated sensitivity of middle-layer MLPs could also stem from amplification, semantic routing, or residual-stream integration mechanisms rather than storage itself. Our experiments in **Figure 3, Figure 4, and Table 8** demonstrate that these MLP components exert disproportionately high influence on safety behavior, which makes them effective targets for DAPA’s training-free restoration procedure, but we do not claim they uniquely contain all alignment signals. We will soften the language accordingly and explicitly acknowledge these alternative explanations in the revision.

---

> > ### Author Response · Authors · 2025-11-18
> > **Rebuttal - Continue**
> >
> > > **Reviewer's Comment**: Pairwise difference-based localization is indicative, not definitive...
> >
> > **Response**:  We appreciate the reviewer’s clarification and fully agree that our pairwise causal-replacement analysis identifies influence rather than providing definitive evidence that the gate projection is the unique repository of alignment knowledge. Our intention is not to claim exclusive storage; rather, our experiments show that the gate projection exhibits high causal sensitivity for recovering aligned behavior under shadow-alignment settings. **Figures 3 and 4** illustrate that modifying the gate projection produces disproportionately strong shifts in harmfulness and refusal behavior, and **Table 8** further confirms that different projection components yield markedly different safety–utility outcomes. These results suggest that the gate projection is a high-leverage locus for restoring safety, not that alignment is stored only there. We will revise the text to explicitly frame the findings as influence-based evidence and acknowledge that fully establishing the underlying representational mechanism would require additional, orthogonal analyses beyond the scope of this work.
> >
> > As a consequence, once the central assumption “alignment is stored in the gate projection” is weakened, the methodological foundation of the paper becomes less solid.
> >
> >
> > > **Reviewer's Comment**:  Another issue with this paper is that the writing is not sufficiently rigorous. There are grammatical errors, ambiguities, logical breaks, and even simple typos ....
> >
> > **Response**: We appreciate the writing-rigor feedback and have corrected the noted issues in the revised version.

---

> > > ### Author Response · Authors · 2025-11-25
> > >
> > > Dear Reviewer,
> > >
> > > We sincerely appreciate your thoughtful feedback. As the discussion period reaches its final week, we would be grateful to know whether our latest responses have resolved your concerns or if any questions remain. We are happy to provide further clarification as needed. Wishing you a Happy Thanksgiving.
> > >
> > > Thanks,
> > >
> > > Authors

---

> > > > ### Author Response · Authors · 2025-11-28
> > > >
> > > > Dear reviewers,
> > > >
> > > > As the rebuttal deadline is approaching, I would like to kindly check whether our responses have addressed your concerns and questions. If there is anything that is still unclear or could benefit from further clarification, please let us know, and we will be happy to elaborate.
> > > >
> > > > Thank you very much for your time and effort in reviewing our work.
> > > >
> > > > Happy Thanksgiving!

---

### Official Review · Reviewer_vKnE · 2025-11-01

**Soundness:** 3
**Presentation:** 3
**Contribution:** 3
**Rating:** 4
**Confidence:** 3

**Summary:**

The paper introduces DAPA (Decoupled Alignment for Robust Plug-and-Play Adaptation), a training-free method that enhances safety in large language models by transferring alignment knowledge from a well-aligned “teacher” to a target model of the same family. It identifies that alignment signals concentrate in MLP gate layers and selectively edits only a small portion of them using a delta-debugging search, effectively restoring safety without retraining. Tested on 17 models including Llama, Mistral, and Gemma, DAPA significantly boosts defense success rates while maintaining model utility, demonstrating an efficient and scalable way to repair “shadow-aligned” models.

**Strengths:**

Introduces training-free, plug-and-play alignment via targeted memory transplantation, a novel framing that removes retraining cost and offers a new perspective on alignment localization in LLMs

Demonstrates strong empirical rigor. It is tested on 17 models across Llama, Gemma, and Mistral families, with consistent DSR gains and detailed ablations confirming that MLP gate layers encode alignment knowledge.

Presents a transparent workflow (delta-debugging search + layer transplantation), clear evaluation metrics, reproducibility statement, and open-sourced code ensuring interpretability and replication

Offers a practical, compute-efficient alternative to RLHF/DPO alignment.

**Weaknesses:**

DAPA explicitly relies on a pre-aligned teacher and cannot align a model from scratch. This raises questions about its motivation — if obtaining a safe teacher model still requires costly alignment methods such as RLHF or DPO, the overall cost advantage may diminish. Moreover, identifying alignment-critical MLP components through delta-debugging also incurs inference cost, yet the paper does not quantify it. It would be helpful to report the actual computational cost of the search phase and compare it directly with SFT, DPO, and RLHF training costs.


The method modifies a relatively large proportion of parameters (6.26% on average), which is much higher than typical adapters like LoRA. My intuition is that a smaller parameter change would better preserve the original model’s behavior, though potentially with weaker safety gains. It would strengthen the paper to show a trade-off curve between the percentage of parameters edited and both safety and utility performance.


In direct comparisons, DAPA’s Defense Success Rate (DSR) remains below that of RLHF and DPO. It would help to clarify whether the reported computational savings (e.g., one hour on a single A100) justify this performance gap in practical deployment scenarios.

The paper does not include a standard supervised fine-tuning (SFT) baseline, which would provide an essential reference point for understanding DAPA’s effectiveness relative to a simpler and inexpensive alignment method. Including SFT results under the same experimental settings would make the comparison more complete.

**Questions:**

Since DAPA requires a pre-aligned teacher model, how should readers interpret the claimed “training-free” benefit? If the teacher itself was aligned via RLHF or DPO, does DAPA still provide a net cost reduction in realistic pipelines?

The paper reports that about 6.26% of parameters are modified on average. Have you evaluated smaller-scale edits (e.g., 1–3%) to see how DSR and task performance degrade?

You compare DAPA with RLHF and DPO but not standard SFT. Could you add SFT results using the same HarmBench or AdvBench setup to contextualize whether DAPA still provides a meaningful gain over simple supervised fine-tuning?

---

> ### Author Response · Authors · 2025-11-18
>
> > **Reviewer's Comment**: DAPA explicitly relies on a pre-aligned teacher and cannot ....
>
> **Response**: Thank you for the opportunity to classify this problem. We agree that DAPA assumes access to a pre-aligned teacher, but this cost is already incurred in practice because downstream models are typically adapted from such RLHF/DPO-aligned checkpoints, and shallow alignment during task-specific SFT often weakens their safety. Our goal is not to replace the initial alignment pipeline but to restore safety after downstream fine-tuning without retraining, which preserves the original cost advantage. We also report the inference-only cost of our delta-debugging search in **Appendix H.15**, and now include the SFT baseline cost (**4h 40m on 4×A100**), demonstrating that DAPA’s search is orders of magnitude cheaper than training-based methods like SFT, DPO, or RLHF.
>
> > **Reviewer's Comment**: The method modifies a relatively large proportion of parameters ... Have you evaluated smaller-scale edits (e.g., 1–3%) to see how DSR and task performance degrade?
>
> **Response**: We appreciate the concern regarding the proportion of edited parameters. As shown in **Tables 10 and 11**, we already report performance across different edit magnitudes, and to further clarify this trade-off, we will include a new figure visualising how safety and utility vary as a function of the percentage of modified MLP components. These results consistently show that while larger edits yield stronger safety recoveries, even smaller edits preserve the model’s original behaviour while still providing meaningful safety gains. In order to more clearly illustrate how parameter changes influence the performance of DAPA, we plot a line chart in **Figure 7**. The figure shows that as the parameter change increases from **0% to 2%**, the DSR varies by approximately **10%**. Beyond this range, further parameter changes have a diminishing impact on model safety performance, indicating that DAPA remains stable even with larger parameter adjustments.
>
> > **Reviewer's Comment**: In direct comparisons, DAPA’s Defense Success Rate (DSR) remains below that of RLHF and DPO.....
>
> **Response**: We appreciate the opportunity you've given us to clarify this issue. While RLHF and DPO achieve higher absolute DSR, these methods require many hours to days of multi-GPU training, whereas DAPA completes in roughly one hour on a single A100, enabling fast, low-cost safety recovery after downstream fine-tuning. In practical deployment settings—where models repeatedly undergo task-specific SFT and experience shallow alignment—this rapid post-hoc correction is often more feasible than re-running full RLHF/DPO pipelines. Thus, DAPA is designed to be a lightweight, practically deployable complement rather than a replacement for heavy training-based alignment. To demonstrate performance in practical deployment scenarios, we conduct an additional experiment (**Appendix I**) under a limited-resource setting (**one hour on a single A100 GPU**). The results show that FROST achieves better performance than traditional alignment methods under the same constraint.
>
> > **Reviewer's Comment**: The paper does not include a standard supervised fine-tuning (SFT) baseline, ... Could you add SFT results ....
>
> **Response**:  We appreciate the chance to work on this issue. We agree that including an SFT baseline strengthens the comparison, and we have now added SFT results under the same experimental conditions. We have added an SFT baseline under identical settings. As shown in **Appendix H.15**, SFT requires **4 hours and 40 minutes on 4×A100 GPUs** yet still achieves a lower DSR (**45.7%**) than DAPA and slightly degrades utility performance, confirming that DAPA offers a better safety–utility trade-off at lower cost. We also find that SFT reduces the reasoning ability of previously unaligned models, especially those originally trained or aligned with DPO, RLHF, or SFT in reasoning-focused domains.
>
> > **Reviewer's Comment**: Since DAPA requires a pre-aligned teacher model, ....
>
> **Response**:  Thank you for allowing us to look into this problem. Our “training-free” claim refers to the fact that DAPA requires **no additional supervised fine-tuning** or RLHF/DPO during the safety-restoration stage; it operates solely via **inference and plug-and-play distillation.** In realistic pipelines, the pre-aligned teacher is already available because downstream models are typically adapted from RLHF/DPO-aligned checkpoints, and shallow alignment during task-specific SFT weakens their safety. Thus, DAPA provides a net cost reduction by enabling rapid, post-hoc safety recovery without re-running expensive RLHF/DPO procedures.

---

> > ### Author Response · Authors · 2025-11-25
> >
> > Dear Reviewer,
> >
> > We sincerely appreciate your thoughtful feedback. As the discussion period reaches its final week, we would be grateful to know whether our latest responses have resolved your concerns or if any questions remain. We are happy to provide further clarification as needed. Wishing you a Happy Thanksgiving.
> >
> > Thanks,
> >
> > Authors

---

> > > ### Author Response · Authors · 2025-11-28
> > >
> > > Dear reviewers,
> > >
> > > As the rebuttal deadline is approaching, I would like to kindly check whether our responses have addressed your concerns and questions. If there is anything that is still unclear or could benefit from further clarification, please let us know, and we will be happy to elaborate.
> > >
> > > Thank you very much for your time and effort in reviewing our work.
> > >
> > > Happy Thanksgiving!

---

### Author Response · Authors · 2025-11-18
**General Rebuttal/Revision Response**

We sincerely thank the reviewers for their insightful feedback and constructive suggestions, which have greatly improved the technical rigor and clarity of our work.

**This paper proposes a training-free alignment framework that restores global safety behavior in downstream-tuned LLMs via memory editing (Sec. 3 & 4)**. Below, we summarize the key strengthened contributions and major revisions incorporated during the rebuttal.

---
## Key Contributions:
-   **New alignment paradigm:** We introduce DAPA, a _training-free_ framework that restores safety in downstream-tuned models suffering from **shallow alignment**, where task-specific SFT weakens preexisting ethical behaviors.

-   **New mechanism of safety transfer:** We leverage **memory editing and delta debugging** to locate and transplant alignment-critical modules from well-aligned teachers to safety-degraded counterparts **(§3.2, §4)**.

-   **New interpretability insight:** Using ROME-style causal tracing, we show that safety-relevant information is **highly likely concentrated in mid-layer MLP gate projections**, offering an interpretable perspective on alignment degradation and recovery **(§5.1–§5.2)**.

-   **New efficiency advantage:** DAPA achieves **comparable defense success** to RLHF/DPO at only **~1 GPU-hour on a single A100**, avoiding costly retraining while preserving downstream task utility—demonstrating practical scalability for real-world post-alignment correction.

---
## Major Revisions:

-   **Contribution Clarification:** Clarified DAPA’s goal as a _training-free method_ to restore safety in models affected by **shallow alignment**, focusing on post-hoc safety recovery rather than full re-alignment. `All Reviewers`
-   **Revised Sec. 2:** Added comparisons with **Model Surgery** (Wang et al., 2024) and **DELMAN** (Wang et al., 2025), highlighting DAPA’s role as a global safety recovery approach rather than fine-grained editing. `Reviewer PhuW`
-   **Revised Sec. 1:** Clarified the challenge and motivation of **shallow alignment**, emphasizing how downstream SFT weakens pre-aligned safety behaviors and motivates DAPA as a post-hoc recovery solution. `Reviewer yRkn`
-   **New Experiments:**
    -   **New Appendix H.15 & Table 19:** Added computational cost comparison with SFT, DPO, and RLHF, showing DAPA’s efficiency. `Reviewer vKnE`
    -   **New Appendix I & Table 21:** Added performance comparison under limited computational resources. `Reviewer vKnE`
    -   **New Appendix J & Table 22:** Added experiments on modern reasoning models (Qwen3, LLaMA3) demonstrating scalability and safety gains. `All Reviewers`
    -   **New Appendix H.16 & Table 20:** Added attention-module analysis to examine its influence on safety alignment. `Reviewer yRkn`
	  -   **New Figure 11:** Added analysis showing the impact of parameter size on model performance, illustrating how DAPA scales across different model capacities.  `Reviewer vKnE`
## Minor Revisions
-   **Revised Sec. 5.3:** Specified that **Llama Guard 2** is used as the Guardrails baseline for fair comparison and clarified its configuration. `Reviewer PhuW`
-   **Revised Abstract:** Corrected grammatical errors and improved clarity for readability and precision.' `Reviewer yRkn`
-   **Removed duplicate words in Sec. 1:** to improve clarity. `Reviewer yRkn`
-   **Corrected spelling error:** in the Figure 2 caption (“Critial” → “Critical”). `Reviewer yRkn`
---
We once again thank the Area Chair and reviewers for their valuable efforts and thoughtful comments. All revisions have been incorporated into the updated manuscript, with changes highlighted in blue.

---

> ### Author Response · Authors · 2025-11-18
> **Common Questions**
>
> > **Reviewer's Comment**:  Questionable Motivation / Unclear Problem Setting
>
> **Response:** We appreciate the reviewer’s observation and agree that the motivation requires clearer articulation. DAPA is designed for the _post-alignment degradation_ scenario—when a **well-aligned model** (e.g., RLHF/DPO-aligned) undergoes **downstream SFT or task adaptation** and consequently loses part of its safety alignment, a phenomenon we term **shallow alignment**. In such cases, practitioners cannot simply revert to the original aligned checkpoint, as it lacks the specialized capabilities introduced during downstream tuning. DAPA offers a **training-free, plug-and-play** solution to _restore_ alignment without re-running costly RLHF/DPO pipelines, which is especially practical under constraints such as **licensing restrictions**, **partial checkpoint access**, or **on-device fine-tuning environments** where retraining is infeasible.
>
> > **Reviewer's Comment**:  **Over-strong Core Assumption about Ethical Knowledge Representation**
>
> **Response:** We thank the reviewer for this insightful comment and agree that ethical or alignment signals may be distributed across multiple architectural components, including attention heads, MLP–attention interactions, and residual connections, rather than being confined to neurons. Our method does not assume that alignment knowledge is exclusively stored in neurons; instead, it uses MLP modules as an empirically validated intervention point. As shown in **Figures 3 and 4 and Table 8**, modifying a small subset of MLP (especially gate) projections produces strong causal effects on safety behavior, suggesting high influence but not exclusive storage. To further validate this, we have added a new experiment in **Appendix H.16** analyzing the attention module’s contribution to safety alignment. The results confirm that while attention also plays a role, MLP layers exert a consistently stronger and more direct influence. We will revise the text to clarify that DAPA leverages these empirical findings in a mechanism-agnostic manner, without assuming any single storage locus for alignment knowledge.

---

### Author Response · Authors · 2025-11-23
**Friendly Reminder**

Dear Reviewers,

We sincerely appreciate your thoughtful feedback. As the discussion period reaches its midpoint, we would be grateful to know whether our latest responses have resolved your concerns or if any questions remain. We are happy to provide further clarification as needed. Wishing you a Happy Thanksgiving.

Thanks,

Authors

---

### Author Response · Authors · 2025-12-02

Dear Chairs,

We sincerely thank the area chair and all reviewers for the time and effort dedicated to evaluating our submission.

The paper initially received four reviews with scores of 2, 4, 4, 6, and after we addressed all questions and concerns during the rebuttal phase, the ratings improved to 4,4,4,6, with all finalizations occurring at least one day before the information leak.

We describe how our [rebuttal](https://openreview.net/forum?id=L49qTH31Bq&noteId=7V6Xi5bxGP) contributed to these outcomes.

Thank you for your consideration.

Authors

---

### Note · Authors · 2026-01-03

I have read and agree with the venue's withdrawal policy on behalf of myself and my co-authors.